# Natural Product Co-Metabolism and the Microbiota–Gut–Brain Axis in Age-Related Diseases

**DOI:** 10.3390/life13010041

**Published:** 2022-12-23

**Authors:** Mark Obrenovich, Sandeep Kumar Singh, Yi Li, George Perry, Bushra Siddiqui, Waqas Haq, V. Prakash Reddy

**Affiliations:** 1Research Service, Department of Veteran’s Affairs Medical Center, Cleveland, OH 44106, USA; 2Department of Chemistry, Case Western Reserve University, Cleveland, OH 44106, USA; 3The Gilgamesh Foundation for Medical Science and Research, Cleveland, OH 44116, USA; 4Department of Medicinal and Biological Chemistry, College of Pharmacy and Pharmaceutical Sciences, University of Toledo, Toledo, OH 43614, USA; 5Departments of Chemistry and Biological and Environmental Sciences, Cleveland State University, Cleveland, OH 44115, USA; 6Indian Scientific Education and Technology (ISET) Foundation, Lucknow 226002, India; 7Department of Nutrition and Dietetics, Saint Louis University, Saint Louis, MO 63103, USA; 8Department of Neuroscience Developmental and Regenerative Biology, University of Texas, San Antonio, TX 78249, USA; 9School of Medicine, Northeast Ohio College of Medicine, Rootstown, OH 44272, USA; 10School of Medicine, University of Pittsburgh, Pittsburgh, PA 15260, USA; 11Department of Chemistry, Missouri University of Science and Technology, Rolla, MO 65409, USA

**Keywords:** natural products, *Ginkgo biloba*, vascular dementia, Alzheimer’s disease, microbiota, microbiome, microbiota–gut–brain axis, metabolic interactome, blood–brain barrier, aging

## Abstract

Complementary alternative medicine approaches are growing treatments of diseases to standard medicine practice. Many of these concepts are being adopted into standard practice and orthomolecular medicine. Age-related diseases, in particular neurodegenerative disorders, are particularly difficult to treat and a cure is likely a distant expectation for many of them. Shifting attention from pharmaceuticals to phytoceuticals and “bugs as drugs” represents a paradigm shift and novel approaches to intervention and management of age-related diseases and downstream effects of aging. Although they have their own unique pathologies, a growing body of evidence suggests Alzheimer’s disease (AD) and vascular dementia (VaD) share common pathology and features. Moreover, normal metabolic processes contribute to detrimental aging and age-related diseases such as AD. Recognizing the role that the cerebral and cardiovascular pathways play in AD and age-related diseases represents a common denominator in their pathobiology. Understanding how prosaic foods and medications are co-metabolized with the gut microbiota (GMB) would advance personalized medicine and represents a paradigm shift in our view of human physiology and biochemistry. Extending that advance to include a new physiology for the advanced age-related diseases would provide new treatment targets for mild cognitive impairment, dementia, and neurodegeneration and may speed up medical advancements for these particularly devastating and debilitating diseases. Here, we explore selected foods and their derivatives and suggest new dementia treatment approaches for age-related diseases that focus on reexamining the role of the GMB.

## 1. Introduction

Natural alternatives are being explored for high-risk medications treating age-related diseases. Age-related neurodegenerative diseases have few treatments and no effective cures to date. During aging, several diseases become more prevalent. Alzheimer’s disease is the most common neurodegenerative disease and vascular dementia is the second most common related dementia in America and their prevalence is increasing in other developed countries [1,2]. Due to its protracted and progressive nature, AD and other neurodegenerative conditions increase with age [3]. AD is a devastating disease of the elderly, which begins before prodromal signs, leads to a decline in cognitive and behavioral capacity, dwindles functional ability, hampers independence, and eventually results in death. The gut microbiota, prebiotics, probiotics, symbiotics, nutritional, or other approaches have the potential for modifying recalcitrant and debilitating conditions and may help diminish the deleterious effects of aging [4]. AD leads to progressive cognitive and behavioral decline in the elderly and with advancing age. Although there are treatments, the disappointing fact is that there are no real promising cures on the immediate horizon and thus, it is extremely important to focus on basic research and novel evidence-based therapeutics with proven potential benefits to delay the onset of this protracted disease. Alzheimer’s Disease International [5,6] estimates that the number of people living with dementia worldwide will increase to 65.7 million by 2030 and 115.4 million by 2050 [7] provided we live into later decades of life. DeKosky and colleagues estimated that even a delay of disease onset by 5 years could reduce the number of dementia cases by approximately 50% after several decades [8,9]. Nevertheless, DeKosky and colleagues did not find, in their work with Ginkgo 120 mg alone, which was less than the 240 in other trials, that it reduced cognitive decline in those aged 72–96 years of age [10]. However, even the slightest progress toward new treatments or discovery would be a welcome advance and one must not exclude the role of the MGB in brain health.

We know that the multifactorial nature of cognitive decline and AD or VaD from our work and that of many others can involve mitochondrial dysfunction and propagators, such as cytokine stress or oxidative stress in the cardiovascular system, brain, and CNS, including brain microvasculature and parenchyma, which results in the accumulation of damaging reactive oxygen species (ROS) [11] and reactive nitrogen species (RNS) in the aging brain [12]. These reactive species in cerebrovascular and other diseases promote leukocyte and immune cell adhesion and increase endothelial and other cellular compartment permeability, which results in chronic injury stimulus cascading in a host of downstream events [12]. One event is mitochondrial damage, which feeds forward oxidative stress and other propagators of downstream chemical modification [11,13]. Other cascades include changes in mitochondrial enzymes in the vascular wall and in brain parenchymal cells and imbalances in the activity of vasoactive substances, such as different isoforms of nitric oxide synthase, endothelin-1, various oxidative stress markers, and mitochondrial DNA damage [12,14].

When damage occurs to the vascular networks of the brain, oxygen and nutrients are no longer able to provide sustenance to vital cells in the brain, including glial cells and neurons. This damage rearranges the natural supporting framework for neurons and associated cells, causes irreversible neuronal changes, leads to dysfunction in the cerebrovascular networks, and eventually contributes to vascular dementia [15]. It is this vascular pathology that causes the detrimental effects in vascular dementia and, given that it often coexists with AD, targeting treatments at the vascular level may help ward off dementia and AD. Progressive cognitive decline and eventual death is still unavoidable at this time, once the disease has begun its course. Considering how much these deleterious changes affect a person’s life and that of the caregiver, the urgency to find an effective treatment is essential and not to be understated. However, as the pathogenesis of AD is still not certain, research is ongoing for new ways to tackle this disease, and, as frightening as it may be, there are few promising treatments or cures to date for AD and related dementias. Animal studies centering on tolerance to hypoxia and aglycemia have revealed that neuroprotection mechanisms through hibernation for AD and VaD are possible [16]. Perhaps hibernation mimetics could be established to offer the same protection in human subjects.

In examining the link between the heart, brain, and the vasculature, we find common potential treatment targets for age-related diseases. When we add the role the microbiota–gut–brain axis plays, as both a novel treatment target and as a contributor and metabolic modulator of disease, we build a case for a new physiology of the elderly. The elderly often cannot tolerate or metabolize drugs well, so we consider safe, natural, inexpensive, and effective alternatives to pharmaceuticals and current dementia treatments. If one turns to alternative treatments such as natural products, prosaic foods, exercise, and even higher education, we find these modalities are associated with protection against AD [17]. Prosaic foods and natural products have been used to treat age-related diseases in ancient and recent medicine. One natural product that has had a lot of interest is *Ginkgo biloba* (*G. biloba*). *Ginkgo biloba*, and its main constituent compounds, the ginkgolides, have become something of a celebrity as far as natural drug candidates go. This treatment strategy stems from extensive research that supports the neuroprotective properties of the ginkgolides and their ability to enhance cognitive function and show promise against anxiety in dementia and other psychiatric disorders [18]. The role of the gut microbiota in cardiovascular disease and aortic stiffness is highlighted by trimethylamine-N-oxide, or TMANO, which is linked to specific microbiota and induces aortic stiffening and increases systolic blood pressure with aging in mice and humans in atherosclerosis [19]. TMANO is an oxidation product of trimethylamine, which is transformed from the metabolism of choline by the gut microbiota through dietary phosphatidylcholine. TMANO plasma concentrations depend on the gut microbial composition, but mostly on diet and the activity of the liver flavin monooxygenase [20,21,22].

## 2. Pathogenesis of Alzheimer’s Disease/Vascular Dementia and Effects of *Ginkgo biloba*

Alzheimer’s disease is characterized by the formation of neurofibrillary tangles [23] and the accumulation of abnormal amyloid beta (Aβ) peptide or smaller soluble Aβ oligomers and their deposition in senile and neuritic plaques [24] and in brain parenchyma and cerebral capillaries [25]. AD is largely a protracted disease, but the exact etiology is largely unknown. Nevertheless, age is implicated to be the strongest risk factor and the presence of a first-order family member also increases risk significantly for developing AD [9]. 

However, these hallmark lesions are mostly end-stage changes and their protracted nature is indicative of the disease, but are not sufficient and necessary for dementia to occur, which suggests earlier mechanisms are involved well before any prodromal stages [17,26]. The role of natural compounds such as *Ginkgo biloba* on immunomodulation, inflammation, and cardiovascular homeostasis have been demonstrated to include the antagonist activity of the platelet activating factor, by reducing the infiltration of eosinophils and neutrophils and alleviating neuroinflammatory injury by inhibiting astrocytic Lipocalin-2 expression after ischemic brain injury and stroke [27]. Taken together, this provides more evidence to support AD as a vascular disorder [28]. 

One of the major potential therapeutic indications for the experimental use of *Ginkgo biloba* extract EGb761 is its ability to protect against Aβ-induced neurotoxicity and neuroinflammation, a key to pathology in Alzheimer’s disease, which also includes cerebral amyloid angiopathy [29]. Moreover, there is a vascular and cerebrovascular–brain axis in AD including VaD since vascular dementia and cardiovascular disease implicate shared vascular mechanisms in the development and/or progression of AD [30,31,32,33]. In the development of this disease, the Aβ deposition in the brain causes a reactionary stimulation of the body’s defense system. In this defense, resident microglia/infiltration macrophages become activated to clear the deposition and concomitantly produce a range of inflammatory mediators that are potentially neurotoxic, which contributes to the pathogenesis of AD [34,35,36]. In a study analyzing the abnormal amyloid beta peptide formation, it was found that *Ginkgo biloba* has an anti-amyloid aggregation effect due to its effect on increasing transthyretin RNA levels, an amyloid beta peptide transporter that decreases amyloid brain deposition [9]. Moreover, amyloid oligomers and downstream fibril formation can be affected or arrested in many ways that are not immediately evident in a clinical study, due to our inability to image such changes in vivo [37]. Many studies in rats and in vitro reveal an alternate method of fighting Aβ accumulation by modulating alpha-secretase, the enzyme responsible for cutting the amyloid beta peptide, and preventing the final fragments from being made [38]. Further, in human subjects, *Ginkgo biloba* was found to have utility in the treatment of cerebral insufficiency due to inflammatory neurotoxicity-induced dementia [39], whether due to degenerative processes or vascular insufficiency [40]. 

Another hallmark lesion in AD is the hyperphosphorylation of the microtubule-associated protein tau [41]. To mimic AD-like pathological conditions and memory deficits, hyperhomocysteinemia (HHcy) rat models were employed. Low folate raises blood HHcy concentrations, which are associated with poor cognitive performance in the general population. Elevated HHcy has neurotoxic and vasotoxic effects in AD and dementia and can be considered an early marker of cognitive decline [42]. HHcy is implicated in many age-related diseases, including neurologic diseases AD and VaD, cardiovascular disease, and ocular disorders such as macular degeneration and diabetic retinopathy, but has a less defined role in vascular dementia. Further, HHcy induces inflammatory-mediated blood–brain barrier (BBB) dysfunction through the activation of glutamate receptor N-methyl-d-Aspartate receptor [43]. HHcy-induced brain inflammation is a mechanism of BBB dysfunction and other immune-privileged barriers [43]. HHcy can also induce oxidative stress, which contributes to endothelial and arterial damage, increases tau microtubule-associated protein phosphorylation, and promotes blood clot formation, all of which are comorbidities of aging. While these models do not result in senile plaque deposition or intracellular inclusions or recapitulate the human condition, rodent models are the best approximation for in vivo changes. DNA damage with apoptosis is another protracted hallmark of AD pathobiology [44,45,46].

A hallmark lesion in AS is the hyperphosphorylation of tau protein. To explore the phosphorylation condition, rats were injected with EGb761 or saline as a sham control [47] and the status of oxidative damage, spatial/learning memory, level of memory-related proteins, tau phosphorylation, level/activity of tau kinase (GSK-β), and protein phosphatase 2A (PP2A) were all measured and reported. This particular group [47] showed that EGb761 could significantly antagonize HHcy-induced oxidative damage and recover PP2Ac and GSK3 β activities deregulated by HHcy. Not only did EGb761 protect against Aβ-induced neurotoxicity, but this study gave promising results with EGb761 for treating AD by decreasing hyperphosphorylation and demonstrating antioxidative activity [47]. When analyzing the abnormal amyloid beta peptide formation, it was found that *Ginkgo biloba* has an anti-amyloid aggregation effect due to its ability to increase transthyretin RNA levels, an amyloid beta peptide transporter that decreases amyloid brain deposition [9,23]. In vitro, *Ginkgo biloba* extract (GLE) provided protection from abnormal Aβ peptide accumulation with deposition in the brain parenchyma and cerebral capillaries [25]. However, native amyloid precursor protein is not sufficient or necessary for AD pathology alone [48] and the microtubule-associated protein tau is another mechanism believed to cause AD pathology [49]. The effects of *Ginkgo biloba* on beta secretase or the cholinergic system as well as increasing neurotransmitters should be further explored as they are both associated with AD [50,51]. In that regard, in studies aimed at improving cognitive function and control, *Ginkgo biloba* extracts were found to increase levels of acetylcholine, dopamine, and 5-hydroxytryptamine as well as to inhibit the activation of acetylcholinesterase, which is responsible for degrading acetylcholine [52]. The use of cholinergic agents, such as acetylcholinesterase (AChE) inhibitors, is currently being explored and these agents have shown efficacy and considerable benefit in AD and VaD therapy [53]. AChE inhibitors include donepezil, rivastigmine, and galantamine, as well as active compounds from the herb huperzine [53]. This accumulating evidence reveals that cholinergic deficiency contributes to vascular dementia and disease, which can be modulated by drugs such as *Z-ligustilide* [54,55]. Other natural products such as Hwangryunhaedok-tang (HT) [56], black tea, green tea, and coffee contain compounds that inhibit acetylcholinesterase and green tea metabolites also inhibit beta secretase, which could prevent the release of Aβ peptides from amyloid precursor protein. However, coffee is a less effective inhibitor of acetylcholinesterase, having no butyrylcholinesterase- or beta secretase-inhibiting activity [57]. 

The popularity for the use of *Ginkgo biloba* stems in part from the finding that it largely has no major adverse events or reactions or known side effects, to date, other than those due to the antagonism of clotting mechanisms, which should be a contraindication in some trials. Moreover, one study showed that when compared to donepezil monotherapy, the adverse event rate was lower in the *Ginkgo biloba* group and even under the combination group when compared to a placebo [58]. Contrary to the expectation that taking two drugs simultaneously would increase adverse events, this study had the opposite effect. In fact, improvement in adverse reactions with donepezil was observed when taken in conjunction with *Ginkgo biloba* extracts. These findings would be expected to lead to renewed interest in this area toward developing Ginkgo-derived compounds as therapeutics for the treatment of dementia. Nutrition companies looking towards commercializing this new discovery can help fund clinical trials and help this newly developed research accumulate more evidence-based research to support the claims of treatment success. 

Amyloid β deposition is associated with the pathogenesis of AD, in particular tissue damage in the brain and associated parenchymal cells and neurons. Oxidative stress and neuroinflammation are associated with the damage caused by this pathology, which contributes significantly to dementia and the deleterious symptoms of AD. One of the cytokines found to increase with AD is CD38. This cytokine, expressed by neurons, astrocytes, and microglial cells, is responsible for regulating the repair and inflammatory processes within the brain [24], among other actions, degrades nicotine adenine dinucleotide (NAD), and regulates the migration of inflammatory cells during neuroinflammation [59,60]. One study examined CD38 expression on AD pathology in CD38-deficient mice and in vitro cultures. All treatments decreased secretions of Aβ from neuronal cultures, decreased the activity of β- and γ-secretase, the enzymes that cleave amyloid precursor protein, and significantly reduced Aβ plaque load and levels of insoluble Aβ [24]. This resulted in the attenuation of AD pathology and improved spatial learning and learning performance. The neuroprotective effects of CD38 inhibition suggest a novel therapeutic approach for AD with GLE. One mechanism of action for GLE and other flavonoids in AD involves attenuating CD38 neuroinflammation. Studies examining select flavonoids reported that, at low molecular levels, CD38 was inhibited by flavonoids, such as luteolinidin, kuromanin, and luteolin [61]. Other studies exploring kuromanin and luteolin found it could inhibit CD38 directly by affecting NAD levels in glycolysis [62]. Taken together, these findings suggest alternative roles for flavanols, such as those found in GLE, which have alternative mechanisms in modifying the hallmark lesions associated with AD pathology, including effects on neuronal metabolism and glyco-oxidative stress. Moreover, the microbiota also plays a key role in flavanol metabolism and must be considered when assigning any effect of GLE on the experimental model.

Also implicated in AD are mitogen-activated protein kinase (MAPK) signaling pathways. MAPK pathways represent a promising therapeutic target as they are implicated in inflammatory and apoptotic processes during cerebral ischemia and reperfusion injury as well [63,64]. Inhibitors of MAPK pathways are being explored as a therapeutic strategy for ischemic stroke. Moreover, bilobalide, a predominant sesquiterpene trilactone constituent of *Ginkgo biloba* leaves, has been shown to exert powerful neuroprotective properties, which are closely related to both anti-inflammatory and anti-apoptotic pathways [63]. This group investigated the neuroprotective roles of bilobalide in middle cerebral artery occlusion and reperfusion and oxygen–glucose deprivation and reoxygenation models of cerebral ischemia/reperfusion injury. They attempted to confirm the hypothesis that the protective effects of bilobalide were through the modulation of pro-inflammatory mediators and MAPK pathways. Their data indicated that the neuroprotective effects of bilobalide on their cerebral injury model are associated with the inhibition of pro-inflammatory mediator production and the downregulation of JNK1/2 and p38 MAPK activation [63].

## 3. Safety of *Ginkgo biloba*, GLE Extracts, and Similar Compounds

As the oldest extant deciduous tree, *Ginkgo biloba,* with its bilobed fan-shaped leaves, is known for its beauty, longevity, and resistance to pathogens [65]. In terms of longevity, it can live for over 1000 years [66]. *Ginkgo biloba* has a rich history for medicinal use dating back 5000 years [65]. Currently, it is a treatment option for acute ischemic stroke in China [39]. It was not until the mid-1960s that it was introduced into Western medicine by German physician–pharmacist Dr. Willmar Schwabe [65]. Standardized *Ginkgo biloba* leaf extract (EGb761) is typically prepared from whole dried green leaves [65], which contain many bioactive compounds. Two main components stand out and are largely responsible for GLE’s putative pharmacological effects (see Figure 1), namely, flavonoid glycosides and terpene trilactones, which represent 24% and 6% of the overall plant content, respectively [23,39]. Other potentially beneficial small molecules and phenolic acids, including aging, are present in *Ginkgo biloba* extracts, which are useful for the treatment of vascular cognitive impairment, vertigo, tinnitus, early diabetic retinopathy, and senile macular degeneration [39,65]. *Ginkgo biloba* leaf extract is also taken for the treatment of inflammation, asthma, and bronchitis [67,68]. 

An important consideration for all of the superfood-related treatments in this work is bleeding risk, which is often associated with potential adverse interactions between some of the used herbal supplements and analgesic drugs with natural and prosaic foods, derivatives, and some of the so-called superfoods—in particular, ginkgo, garlic, ginger, bilberry, dong quai, ginseng, turmeric, and willow bark [69], as well as with those containing coumarin, i.e., chamomile, horse chestnut, fenugreek, and red clover. That said, taking the herbal or whole food extracts can limit the risk. Nevertheless, the whole food of *Ginkgo biloba* and its GLE extracts together with the interaction of *Ginkgo biloba* with non-steroidal anti-inflammatories could complicate the bleeding risk, since the combined use may induce increased bleeding risk to sensitive patients. However, these risks were not exclusory in the studies elucidated. Further, it is known that the use of the whole *Ginkgo biloba* extract and not simply the fractions can be the better form for consumption due to the synergic effect that occurs when consuming the different fractions together. Interestingly, the hibernating ground squirrel has a defect in clotting, which may offer clues into the neuroprotective mechanisms or mimic natural compounds such as those discussed [70,71].

Commercial *Ginkgo biloba* standardized extracts are available in the US market as coated tablets for oral administration [65] with a recommended daily dose of 240 mg, either taken in one or two doses. One of the most important contraindications when supplementing orally with *Ginkgo biloba* is counter indicated for anyone receiving anticoagulation, antiplatelet therapy, or for those with bleeding disorders [65]. This is due to the ability of *Ginkgo biloba* to form free platelets by antagonizing platelet-activating factors [39] and inhibiting platelet aggregation [65]. Administration is contraindicated in children and during pregnancy and nursing because of the potential adverse effects [65].

The safety and efficacy of ginkgolides in AD and VaD can be found in a *Ginkgo biloba* randomized 400 patient clinical trial of 50 years or older with VaD or AD who were given a special extract EGb761 or placebo for 22 weeks [72]. Patients who scored below 36 on the test for the early detection of dementia, with discrimination from depression, and who also scored between 9 and 23 on the Short Syndrome Test battery and at least 5 on the Neuropsychiatric Inventory were chosen for this particular trial. The drug was well tolerated and the adverse events were no greater than those of the placebo treatment according to this clinical trial, which focused on safety. Moreover, the treatment was found to improve cognitive functioning and behavior symptoms in patients with neuropsychiatric features and age-associated cognitive impairment or mild to moderate dementia. Since the treatment results were essentially similar for the AD and VaD subgroups, the data supports the safety and efficacy of EGb761 in the treatment of cognitive and non-cognitive symptoms of dementia overall, notwithstanding any interference with clotting [72]. A clinical trial examined the efficacy of *Ginkgo biloba* on the neuropsychological functioning of cognitively intact adults aged 60 years or older assigned randomly to receive either 180 mg/day of GLE or a placebo daily for 6 weeks. Compared to the placebo group, the group taking GLE rated their overall abilities as “improved”. This study harmonizes evidence for the potential efficacy of GLE. 

However, the findings on *Ginkgo* are mixed. A larger study of three thousand and sixty-nine community volunteers 75 years or older conducted from a total of five academic medical centers in the United States revealed that 120 mg of GLE twice a day was not effective in reducing the overall incidence of dementia. In this study, five hundred and twenty-three individuals developed dementia with 92% classified as possible or probable AD or VaD with evidence of vascular disease of the brain, offering hope for the vasculopathies in AD and VaD [73]. We do not know enough about why the outcomes were mixed, but poor diet in the elderly and the co-metabolism of the host with select gut microbiota can contribute or modulate the pathogenesis of AD and VaD, which could explain any variant results. Nevertheless, an association was found between gut microbial composition and arterial stiffness and the GM has an influence on the aging of the vasculature [74]. Therefore, one must consider the impact a poor diet has on the gut microbiota composition and overall health in the elderly [75]. It was observed that in the elderly population over 100 years of age, centenarians, the structure of their human gut microbiota greatly differs from that of young adults and people 70 years of age. Centenarians were described as having an increased number of facultative anaerobes leading to an increased inflammatory status [76]. 

## 4. Blood Flow and a Leaky Gut, Leaky Brain in AD, VaD, and Aging

We have shown a clear connection for VaD and AD centering on G-protein-coupled receptor kinase 2 (GRK2) and AD [31], which was the basis for the concept of the heart–brain connection in vascular AD, vascular dementia, and other forms of Alzheimer’s disease [26]. Moreover, there are several metabolite-sensing G-protein-coupled receptors that bind with short-chain fatty organic acids (SCFAs), which are important for overall gut health and immune response regulation [77]. 

The mechanism of action of *Ginkgo* is largely to improve brain perfusion or blood flow. Central to blood flow is the integrity and sensitivity of blood-privileged immune barriers to the environment around it. The BBB and/or gastrointestinal barrier can experience disruption and increased permeability due to surrounding inflammation or other stressors, including oxidative stress, possibly through mitochondrial dysfunction or changes [26,78]. Each immune-privileged barrier is important for the protection of vital organs from pathogens and other assaults; a natural immune barrier maintains this separation from the rest of the body. These immune barriers also function in cellular communication and movement of nutrients and other factors. Moreover, the immune-privileged barriers, including the gastrointestinal tract epithelial barrier and blood–brain barrier, become significantly more permeable with advancing age [79]. As previously stated, this can may make the CNS more susceptible to neurotoxins generated by pathogenic resident or environmental microbiota (45), especially during chronic infection. Aging and inflammation progressively alter BBB permeability and can facilitate increased cerebral viral infection and opportunistic pathogen colonization as we age [80]. This is not surprising given that vascular-type dementias are the second most common form of dementia after Alzheimer’s disease [52,81] and both AD and VaD are age-related diseases. Moreover, cerebrovascular changes and associated amyloid beta peptide deposition in the brain, and elsewhere, are due, in part, to the leakiness of the blood–brain barrier [78,82]. Moreover, the role of the gut in dysbiosis or the production of xenobiotic molecules could affect the CNS and brain at both the blood–brain barrier interface [83] and from gut dysbiosis, which could be a cause and a consequence of increased levels of oxidative stress, since anaerobes thrive in the presence of electron acceptors [84].

The clear role of mitochondria in cerebrovascular hypoperfusion and models demonstrate that dysfunction is characterized by cerebral hypoperfusion and loss of the BBB integrity, and these occur before any signs of frank AD and in the prodromal stages [31]. After this stage, the pathogenesis is associated with increased Aβ deposition in the cerebral vessel walls and elsewhere. Concomitantly, or due to hypoperfusion injury, mitochondrial damage ensues along with increased oxidative stressors and the production of ROS/RNS and the release of mitochondrial apoptotic proteins and consequent induction of brain endothelial cell degeneration and death [85]. These changes contribute to the downstream induction and activation of nitric oxide [86] and of the transcription factor hypoxia-inducible factor-1α [87], which regulates the response to hypoxic conditions, which are involved in amyloid processing [88] and amyloidogenic mechanisms involving amyloid-β precursor protein and eventually aberrant Aβ production [89].

In order to fully understand how *Ginkgo biloba* can play a role in the treatment of AD and VaD, it is important to understand how blood flow and our gut epithelial and blood–brain barriers may be affected. When there is a disruption of blood flow to any area in the body through occlusive ischemia, such as thromboembolism or strangulation, the affected area either perishes or experiences a reintroduction of blood flow when the disruption is corrected. However, this reperfusion causes injury when the reintroduction of blood flow leads to the production of reactive oxygen species (ROS) and reactive nitrogen species (RNS) and free radical formation and propagation. As blood flow decreases in high oxygen-requiring tissues, especially the brain, ROS and RNS accelerate the pathological changes in some diseases including AD [9] and damages protein, nucleic acids, and lipids, particularly the vulnerable hippocampal neurons, the vasculature, and the BBB [78,90,91]. These are all, in part, also involved in the breakdown of the blood–brain barrier and/or gastrointestinal blood barrier, which contributes to the so-called leaky gut and leaky brain [78]. Interestingly, research shows that *Ginkgo biloba* enhances cerebral blood flow [39], scavenges free radicals [92], dilates blood vessels [9], decreases blood viscosity and erythrocyte deformability [9], and can improve BBB integrity, likely by quenching oxidative stress, since *Ginkgo biloba* has demonstrated antioxidant properties [93,94]. Targeting oxidative stress, one of the many components in the dementia-developing brain, suggests additional support for *Ginkgo biloba* by affecting the underlying mechanisms of pathology in AD and VAD. 

During systemic inflammation, often due to infection, not unlike neurologic disease, the changes in the blood–brain barrier include abnormal sensitivity to the effects of systemic inflammation [95]. The effects of systemic inflammation are modulated by cytokines, interleukins (IL), neutrophils, leucocytes, complement cascade activation, antibody complexes, T cells, and/or other cell-mediated immune responses. Together, the gut–blood and blood–brain barriers provide a diffusion-selective gate that helps monitor and prevent the entry of select substances and bacteria into host organs and tissues. Inflammation from cytokine stressors, such as lipopolysaccharide, can eventually affect the integrity of these barriers, which can affect brain health [96,97,98]. For example, interleukin-1 and IL-6 can increase cortisol release by the stimulation of the hypothalamic pituitary arm of the microbiota–gut–brain (MGB) axis, which is often exhibited in stressed or depressed patients [99]. In addition to the microbiome, the mycobiome participates in modulating the cytokine production [100]. Thus, cytokines produced at the gut level do reach the brain via the bloodstream. For example, in autism, it was found some molecules may cross the blood–brain barrier and modulate brain area stimulation [101]. This systemic inflammation has been associated with antibiotic use, depression [97], and psychiatric comorbidities that occur in irritable bowel disease [78]. As the disruption of the blood–brain barrier occurs due to surrounding inflammatory reactions, this so-called leaky brain now leads to increased cerebrospinal fluid (CSF) protein production and its translocation. As the leakiness continues, so does the movement of small molecules, antibiotics, and phagocytes into the brain [78,102]. Also linked to the gut microbiota is their access to peripheral blood, for example during sepsis and infections, which can lead to serious illness. 

## 5. Barrier Breaches and the Microbiota

Natural immune barriers are important for the protection of our brain, where they maintain separation of the immune-privileged environments from pathogens and the rest of the body. With assaults, the BBB can break down or become leaky, resulting in bacterial translocation and infiltration [103]. Vascular-type dementias are the second most common form of dementia after Alzheimer’s disease [52,81]. AD and VaD share cerebrovascular changes and with (Aβ) peptide deposition in the brain. The importance of breaching the tight barriers in the gastrointestinal system and blood–brain barrier is their link to alterations in the gut microbiota and the association with neurodegenerative diseases [104]. We suggest that a leaky gut can predispose the individual to increased blood–brain barrier permeability and promote neuroinflammation. Evidence now suggests this is due, in part, to leakiness of the BBB [78,82]. Further, we have seen the translocation of Aβ, proteins, and cytokines, which are found distal to the brain and, at the same time, can enter the brain to cause damage to neurons through the disrupted BBB [105]. Moreover, alterations or breaches in the gut microbiota composition can increased gut–blood barrier permeability and lead to immune-activated downstream systemic inflammation. Examples of bacteria translocating in AD include *Porphyromonas gingivalis*, which has been observed passing from oral sites of infection via olfactory or trigeminal cranial nerves into the brains of neurodegenerative disease patients [106,107]. This example supports the notion that neurological diseases have a relationship with the microbiota [106]. Moreover, it is not simply bacteria, as the intracellular protozoan, *Toxoplasma gondii*, can cross the BBB and cause neurological dysfunction and encephalitis, which suggests that the microbiota may be influencing neurological function over time to contribute to pathogenesis [78,106]. 

In order to understand how systemic changes in immune-privileged organ breaches occur, we consider mechanisms of host barrier integrity [108]. The gut can influence the blood–brain barrier permeability by means of oxidative stress, cytokine stress, gastrointestinal-derived hormonal secretion, small molecules and metabolic cofactor production, and other inflammatory mechanisms [108]. It was this BBB pathology coupled with the expression of the transmembrane receptor for advanced glycation end-products (RAGE) that a mechanism for translocation and the Maillard reaction was found in AD patients. Moreover, Vienna and colleagues show that the accumulation of AGEs and glycation crosslinking of intimal arterial structural proteins and oxidative stress together with novel mechanisms mediated trimethylamine N-oxide-induced arterial stiffening [19]. For AD pathology, and working through a positive feedback loop, it was discovered that as amyloid beta peptide accumulated, RAGE expression increased, and as RAGE expression increased, so did the breakdown of the BBB’s integrity [25]. Aging and age-related diseases are inextricably linked to metabolism and caloric restriction is the only proven anti-aging intervention to date. What we have not considered fully is the role that the microbiota, also referred to as our second genome, has on our metabolism of superfoods, prosaic foods, or nutrition. The author has long maintained a role for the Maillard reaction, glycation, and oxidation coupled with metal catalyzed oxidation, called glycol-oxidative stress, components in AD pathogenesis [109] and there is now a form of AD called diabetes type IIIc. This form of AD inextricably links metabolism, pancreatogenic diabetes, and the exocrine and digestive properties of the pancreas to age-related diseases. In that regard, we can also add that the microbiota can play a role as part of the host microbiome interaction, namely, the microbiota–gut–brain–endocrine metabolic interactome [99].

*Ginkgo biloba* is gaining interest not only from a nutritional and biochemical perspective, but from a microbiota–gut–brain axis and leaky gut–leaky brain perspective as well. A potential role for *Ginkgo biloba* in RAGE receptor activation and expression and in BBB permeability was illustrated in an in vitro study that mimicked the BBB conditions found in Alzheimer’s disease patients. Examining the effects of EGb761, it was found that the upregulation of RAGE expression was significantly reversed with EGb761 [110]. Further dosing with EGb761 attenuated cell injury, apoptosis, generation of intracellular ROS, BBB leakiness, and RAGE expression [25]. This not only suggests an important progression in the steps to bring about change for Alzheimer’s patients cognitively, but suggests that these modalities may help stabilize the BBB and decrease the accumulation of Aβ peptides in the brain parenchyma. This finding suggests that *Ginkgo* may affect AD and possible type III-c diabetes through the RAGE receptor and may affect the brain, through decreasing mechanisms of glyco-oxidative stress, glycation, or glycotoxins [111]. However, what is not well characterized, and warrants further study, is the effect of prosaic foods and so-called superfoods, and the natural products they represent, such as the effects *Ginkgo biloba* has when co-metabolized within our dynamic gut and its colonizing flora [17].

It is important to note that other prosaic foods and supplements have similar results in vitro. Histopathological alterations that induce deficits in the respiratory chain complex function beget mitochondrial dysfunction, which is intrinsically tied to oxidative stress. Using the selective mitochondrial antioxidants acetyl-L-carnitine and R-lipoic acid, the author published support for improved cerebral blood flow and a trend in improved cognitive function on human ApoE4 transgenic mice and rats in an age-dependent manner [112], and we suggest *Ginkgo* may have similar neuroprotective effects as a phytoceutical and mitochondrial protectant. Although the data are quite heterogeneous, *Ginkgo biloba* extract seems to be of therapeutic benefit in the treatment of mild to moderate dementia of different etiology including mitochondrial dysfunction. These events are found in aging, age-related disease, and in the pathogenesis of aging-related neurodegeneration, and mitochondrial protection and the subsequent reduction of oxidative stress are important components of the neuroprotective activity of GLE extract [39,113]. 

Moreover, when monoclonal antibodies were made against recombinant α-synuclein protein or α-synuclein epitope 118–123 and then applied to 180 prosaic food product antigens, the antibody–antigen reaction and inhibition or cross-reactivity was studied [114]. This group found that food proteins containing peptides most homologous with α-synuclein showed the highest cross-reactivity, which suggests an autoimmune aspect to Parkinson’s disease and likely other brain diseases that involve proteinopathies. Further, molecular similarity with brain antigens involved in synucleinopathy was demonstrated between food peptides with cross-reactive epitopes with human α-synuclein. The converse could also be true for the conformational diseases, where cross-reactivity to components of superfoods could bind and inhibit aberrant forms of proteins and the structural cross-β structure such as amyloid [115] or paired helical filaments [116]. The author suggests this mechanism could be protective for neurodegenerative diseases such as AD and prion disease [117] and suggests that compounds such as curcumin could bind cross-beta or beta-pleated sheet or barrel structures found in many aberrant proteins, which can occur during disease pathogenesis. The natural product or so-called superfood, curcumin, which is a member of the diarylheptanoid compounds, undergoes microbial transformation. Recently, the bioactivity of the isolated metabolites were studied, as they are potentially therapeutic for several diseases, including neurodegenerative diseases [118].

Interestingly, *Ginkgo biloba* shares biologically active secondary metabolites with other prosaic foods. Many compounds isolated from *Ginkgo biloba* were evaluated for their antioxidant activity and compared to ascorbic acid [119]. Ginkgolides and bilobalide (31.3 μg/mL), which are unique constituents of *Ginkgo biloba*, were not very potent inhibitors of the respiratory burst with a range of inhibition between 28% and 37.3%. However, strong antioxidant activity was observed with flavonoids, but not bioflavonoids, and these researchers found that the hydroxyl group in the C ring (3-OH) conferred more antioxidant activity for compounds including kaempferol, quercetin, myricetin, and tamarixetin than if the functional group was absent, e.g., bioflavonoids, apigenin, and 4′-O-methylapigenin. This finding contradicts others, who suggested that antioxidant potency increases with the increasing number of hydroxyl groups [119].

Finally, the coadministration of GLE with other drugs or treatment modalities would be helpful. For instance, in the modulation of this type of diabetes from the standpoint of oxidative stress, improved vascular blood flow and metabolism could be examined with the insulin sensitizer *metformin*, which has been attributed anti-aging and to many positive health effects. In diabetic mice, *metformin* was able to correct memory impairment and the abnormal transport of Aβ across the BBB [120]. Further, GLE (EGB 761) and hyperbaric oxygen therapy, administered together, were shown to ameliorate cognitive and memory impairment in AD via the nuclear transcription factor kappa-B pathway [121]. Another point of the manuscript was to consider the role the microbiota has on oxidative stress and the metabolic products from co-metabolism with the host’s systems, whether diabetic or not.

## 6. Altered Microbiota in the Processing of Prosaic Foods and Antibiotics 

With increased curiosity and movement to discover novel, innovative, and safe alternative treatments for our overall health, including mental and behavioral health, abnormal gut microbiota and/or loss of the intestinal blood barrier are now being linked to abnormalities in memory, depression, mood, anxiety [78,104], and even Alzheimer’s disease [104,122]. In that regard, natural products and so-called superfoods are key targets in future exploration, together with the MGB, for AD and VaD. The importance of this overreaching consideration is to focus on the role of the microbiota in assessing the etiology of a wide range of neurodegenerative and neuropsychiatric diseases and disorders. This support is strengthened by the fact that prosaic foods are co-metabolized with our microbiota together with our own metabolic systems. Moreover, trace amines, 5-hydroxyindoleacetic acid, homovanillic acid, serotonin, and oxitriptan are implicated in this process as they are found in the CNS of said affected patients, which incidentally can also be produced by bacteria [104]. The same holds for phytoceuticals that have suspect affect or putative effects on AD and VaD. 

Moreover, the disruption of the blood–brain barrier allows for microbial access to occur either by transcytosis, crossing microvascular endothelial cells, pericytes, or by parasitosis between cells or by hiding inside a host cell [123], which can increase inflammation. Conversely, of particular interest are microbes that access the peripheral blood or affect the BBB, which can allow amyloid beta proteins and cytokines to pass into the periphery or enter the brain to cause damage to neurons through this disruption [105,124]. Lending support to this link, the study of other neurodegenerative disorders, such as Parkinson’s disease and even autism, has identified microbial-driven components in their pathogenesis within the gut–brain–endocrine interplay [104,125]. For example, in autism, it was found that some compounds do cross the BBB and modulate or stimulate brain areas [101].

Communication between the gut microbiota and the brain is now gaining popularity in its significant role in a broad spectrum of diseases [104]. For example, after microbial colonization with short-chain fatty acid-producing bacteria or after short-chain fatty acid administration or supplementation after *C. difficile* infection in a germ-free animal experiment with rodents, it was found that the presence of tight junctions in the digestive tract of rodents can be improved, thereby implicating the role in strengthening the gastrointestinal barrier [126]. These findings add to a growing body of evidence that show that disruptions in the BBB are linked to a disruption of the gastrointestinal barrier and that probiotics could potentially prevent this disruption by improving BBB integrity and potentially restoring the colonization of microbes that contribute to the barrier [127]. Therefore, one cannot ignore the role for microbial metabolism and biotransformation in modulating *Ginkgo biloba* and any superfood or co-metabolized compounds with the host. Moreover, exploring which microbes are the “right microbes”, or the “eumicrobial milieu”, could improve outcomes for the administration of any natural product or their derivatives.

## 7. Bioavailability, Bioactivity of GLE Components, and Antibiotic Use

*Ginkgo biloba* leaf contains many components, but most of the bioactivity is attributed to the terpene lactones and flavonols. Many constituents are largely unabsorbed in parent forms, such as the terpene lactones [128,129,130], flavonols, and glycosides, some of which must be broken down into readily absorbable compounds. This is where the gut microbiota may play a significant role, providing enzymes such as glycosidase and metabolism to break down the glycosides and release the absorbable forms and aglycones [131,132,133]. 

Our diet, lifestyle, and medications, particularly antibiotics, influence and shape the gut microbiome throughout our life. In that regard, *Ginkgo biloba* is known for its resistance to serious plant diseases and pathogens [134]. The findings show that small amounts of crude GLE extracts (7.8 ug/mL) possess inhibitory activity against Gram-positive and Gram-negative bacteria [134]. One study investigated the antibacterial activity and potency of different solvent fractions from a Himalayan gymnosperm *Ginkgo biloba longa*, collected at high altitude in Kumaun Himalaya, India. The leaf extracts, solvated in methanol, ethanol, chloroform, and hexane, were assessed against the animal and plant bacterial strains *Agrobacterium tumefaciens*, *Bacillus subtilis*, *Escherichia coli*, *Erwinia chrysanthemi*, and *Xanthomonas phaseoli*. The zone of inhibition, minimum inhibitory concentration, and minimum bactericidal concentration values were utilized to quantitatively assess the antibacterial activity and potency of the fractions. The methanol extract showed the highest activity as in zones of inhibition in millimeters, which was 15–21 mm, followed by ethanol and hexanes (14–19 mm) and chloroform (15–20 mm), which indicated that the methanol extract inhibited the growth of all of the bacterial strains. It was also found that all of the GLE extracts were more active than the used standard antibiotics, *erythromycin*, *ampicillin*, and *gentamycin*. Taken together, these findings reinforce the power *Ginkgo biloba* holds in influencing the human microbiota and altering the mycobiome. 

One study aimed at further investigating how fluctuations in the gut microbiota composition were undertaken with *ciprofloxacin*-treated mice to investigate the pharmacokinetics of bioactive extracts and specific terpene lactones including bilobalide, ginkgolide A, ginkgolide B, and ginkgolide C as well as specific *Ginkgo* flavonols, isorhamnetin, kaempferol, and quercetin [135]. The study found that the operational taxonomic unit value and the phylogenetic diversity was lower in the ciprofloxacin-treated mice than in the control mice. Further, the number of total cultured colonies was suppressed by *ciprofloxacin* treatment when mouse stools were cultured in blood, liver, and deoxycholate hydrogen sulfide lactose agar plates. *Ciprofloxacin* administration led to alterations in the pharmacological potency of leaf extracts by diminishing the taxonomic richness after only 5 days of exposure and impacted the levels of about 33% of the bacteria taxa in the gut [136]. In the antibacterial-treated mice, the maximum plasma concentration of isorhamnetin was significantly (*p* < 0.05) increased 1.7-fold in the *ciprofloxacin*-treated mice [135], which indicated that the uptake of isorhamnetin was increased by antibacterial treatment and that the antibacterial consumption may increase the bioavailability of isorhamnetin by suppressing gut microbial metabolic activity. The results indicate that ciprofloxacin administration decreased the number and diversity of the gut microbiota. This indicated that the uptake of isorhamnetin was increased by antibacterial treatment and that the antibacterial consumption may increase the bioavailability of isorhamnetin by suppressing gut microbial metabolic activities. There are other studies into the fluctuations in diversity and enzyme activity of the gut microbiota in response to external factors such antibiotics and probiotics. Another study found that the treatment of rats with antibiotics caused alterations in the metabolic activities of the gut microbiota, which led to alterations of the pharmacokinetics of hesperidin [137]. 

Many other studies have looked deeper into the fluctuations in diversity and enzyme activity the gut microbiota undergoes in response to external factors, such antibiotics and probiotics. Another study looked at *ampicillin*-treated mice to find that alterations of the gut microbiota composition modulated the pharmacokinetics of aspirin in rats [131,138,139] and to their biotransformation into novel or new compounds with co-metabolism. Taken together, the active components of *Ginkgo biloba* leaf extracts, prosaic foods, or natural products undergo biotransformation by the gut microbiota to allow for its absorption through the digestive tract. This study demonstrates that the gut microbes play a role in the biotransformation of orally administered phytochemicals and contributes to the activation or detoxification of phytochemicals that are orally ingested [131]. This finding lends support to the fact that the active components of *Ginkgo biloba* leaf extracts can endure microbial biotransformation in the gut, which allows for metabolite absorption through the digestive tract. It also shows that with antibacterial treatment, changes occur to the GLE extracts, leading to alterations in the pharmacological potency of GLE extracts and to the formation of important secondary metabolites. 

## 8. Comparison of *Ginkgo biloba* to Common Alzheimer Medications

Going head-to-head with the palliative medications currently used in the treatment of dementia, which include memantine, donepezil, galantamine, and rivastigmine [140], is a standard approach. However, there are few drugs that can be considered the gold standard when comparing the effectiveness of natural products and all drugs have drawbacks and exhibit adverse effects; added to that is their high cost. These are factors partly responsible for the popularity of Ginkgo biloba in comparison to currently recommended dementia medications. Considerable overlap occurs between VaD and AD as both show decreases in neurotransmitters in several studies with AD and VaD patients. The significance of neurotransmitter imbalance to the pathobiology of vascular dementia and AD is through cholinergic deficits, which occur in the brain and CSF in both diseases [53]. Injury to cholinergic fibers leads to the decrease in several neurotransmitters [53]. Improving cognitive decline and the symptoms of AD and VaD with cholinergic therapies, including acetylcholinesterase inhibitors, have shown promising effects in patients [141,142]. Acetylcholinesterase inhibitors are currently being explored and have shown promising effects on cognitive improvement in AD and VaD patients [53,143]. Cholinergic agents have been proposed for improving cognitive decline and the symptoms of AD and VaD, including AChE inhibitors as well as natural products such as active compounds from the herb huperzine [53].

If dopamine has a role in neurodegeneration and can covalently modify proteins such as parkin [144], for example, and *Ginkgo biloba* can act at the level of neurotransmission, we could venture to conclude that one mechanism of *Ginkgo’s* action is by increasing dopamine levels. Animal studies with rats led to the discovery that with chronic EGb761 administration, dopamine levels in the prefrontal cortex, a region involved in executive function, memory, intelligence, language, and visual search and gaze control [52,145], were increased by the flavonol glycosides and ginkgolide fractions improved cognitive function [81]. Another study with *Ginkgo biloba* extracts explored improving cognitive function and control by modulating the levels of acetylcholine, dopamine, and 5-hydroxytryptamine (serotonin). These researchers found that *Ginkgo biloba* extracts increased select neurotransmitters, while concomitantly inhibited the activation of acetylcholinesterase, an enzyme responsible for degrading the neurotransmitter acetylcholine [52,142]. The finding supports the case that *Ginkgo biloba* activity and current acetylcholine drugs, used for dementia, both work to increase important neurotransmitters. The accumulating evidence suggests that decreasing cholinergic deficiency can contribute to delayed brain damage in VaD and AD [54,55] and that there is a potential role for *Ginkgo biloba* in this mechanism. The use of ginkgolide and related compounds complements findings centered around the French paradox, as both wine compounds and ginkgolide share mechanistic features in common with age-related brain diseases. Importantly, both *Ginkgo biloba* and red wine constituents have acetyl cholinesterase-inhibiting activity [142]. What is not known is whether there is a role for bacterial co-metabolism and the microbiota–gut–brain axis that may contribute to the beneficial findings [142].

There are several other trials worth mentioning that advocate the success of *Ginkgo biloba* in randomized placebo-controlled double-blind studies (see Table 1), which compare *Ginkgo biloba* (160 mg daily dose) vs. *donepezil* (5 mg daily dose) vs. placebo for 6 months. The results showed that the effect of *Ginkgo biloba* were comparable to donepezil in its improvement of the patients’ attention, memory, and cognitive performance after 6 months of treatment [146]. In addition, another randomized double-blind exploratory trial also compared *Ginkgo biloba* (240 mg/day) vs. *donepezil* (initially 5 mg after 4 weeks 10 mg/day) vs. combined (same doses) for 22 weeks, and the results revealed that there was no difference between taking *Ginkgo biloba* alone or *donepezil* alone and, in fact, combined therapy was superior to monotherapy with *donepezil* because of the decreased side effects and improvement in the patients’ cognitive performance and neuropsychiatric symptoms [58]. Among other clinical trial studies looking at both dementia to due Alzheimer’s or vascular etiology, the multi-center, double-blind, randomized placebo-controlled trials found that treatment with 240 mg of EGb761 daily resulted in significant and clinically relevant improvement in cognition and quality of life of patients and caregivers when compared to placebo [147]. A meta-analysis of eight randomized, double-blind, placebo-controlled clinical trials assessed the efficacy *Ginkgo biloba* in the symptomatic treatment of cerebral insufficiency [39,65]. Treatment with an average dose of 150 mg of EGb761 or placebo for 6 to 12 weeks showed superior results with EGb761 in terms of treating memory deficits, disturbances in concentration, depressive emotional conditions, dizziness, tinnitus, and headache [65]. To further provide credibility to *Ginkgo biloba* and its pharmacological effects, in a study of elderly patients with mild to moderate dementia, the oral administration of EGb761 produced similar EEG profiles of anti-dementia drugs [65]. Moreover, healthy male subjects taking *Ginkgo biloba* exhibited increased alpha activity and cognitive activating-type response, which was more apparent when compared to placebo [65] and demonstrated the credibility and effectiveness of *Ginkgo biloba* when compared to currently used modern drugs, such as *donepezil*. 

*G. biloba* extract is one of the most important and popular dietary supplements that can prevent neurological damage and help in memory enhancement [148]. Previous studies reported the neuroprotective properties of the leaf extract EGb 761 in Alzheimer’s disease at a dose of 240 mg/kg/day [148,149], which may be due to its antioxidant properties and ability to inhibit Aβ-induced neurotoxicity and apoptosis [150,151,152]. Similarly, isolated polysaccharides from *G. biloba* leaf extract were found to be effective against ischemia/reperfusion injury in rat brains when given 7 days before the procedure [153], which resulted in progress in neurological deficits and improved motor function by a reduction in proinflammatory cytokines (TNF-α and IL-1β) and MDA content, while the elevation in the activity of superoxide dismutase, myeloperoxidase, and anti-inflammatory cytokine IL-10 have also been observed (see Figure 2). A *Ginkgo biloba* leaf extract was tested against 6-hydroxydopamine-induced neurotoxicity in rats and was found effective in reducing the behavioral deficit [154]. The underlying mechanism for the neuroprotective effect was highlighted by CA1 neuron protection from death and the downregulation of COX III mRNA encoded by mitochondrial DNA. *Ginkgo biloba* extract was proven to have neuroprotective properties, as well as neurorestorative properties, when studied in mice [155].

**Table 1 life-13-00041-t001:** List of selected clinical trials for the use of natural products of *G. biloba* in the treatment of vascular Alzheimer’s disease, vascular dementia, and AD.

Clinical Trial	Objective	Conclusions
[156]	To assess the efficacy and safety of EGb in Alzheimer’s disease and multi-infarct dementia.	EGb was safe and appears capable of stabilizing and, in a substantial number of cases, improving the cognitive performance and the social functioning of dementia patients for 6 months to 1 year. Although modest, the changes induced by EGb were objectively measured by the ADAS-Cog and were of sufficient magnitude to be recognized by the caregivers in the GERRI.
[157]	This randomized, double-blind, exploratory trial was undertaken to compare treatment effects and tolerability of EGb 761(R), donepezil, and combined treatment in patients with AD and neuropsychiatric features.	These exploratory findings helped to develop three hypotheses that will have to be proven in further studies: (1) there is no significant difference in the efficiency between EGb 761(R) and donepezil, (2) a combination therapy will be superior to a mono-therapy with one of both substances, and (3) there will be fewer side effects under a combination therapy than under mono-therapy with donepezil.
[158]	The purpose of this research was to conduct the first known large-scale clinical trial of the efficacy of *G. biloba* extract (EGb 761) on the neuropsychological functioning of cognitively intact older adults.	Overall, the results from both objectives, standardized, neuropsychological tests and a subjective, follow-up self-report questionnaire provided complementary evidence of the potential efficacy of *G. biloba* EGb 761 in enhancing certain neuropsychological/memory processes of cognitively intact older adults, 60 years of age and over.
[159]	To determine effectiveness of *G. biloba* vs. placebo in reducing the incidence of all-cause dementia and Alzheimer’s disease in elderly individuals with normal cognition and those with mild cognitive impairment (MCI).	In this study, *G. biloba* at 120 mg twice a day was not effective in reducing either the overall incidence rate of dementia or AD incidence in elderly individuals with normal cognition or those with MCI. Trial Registration clinicaltrials.gov Identifier: NCT00010803.
[160]	The aim was to determine the effectiveness and the safety profile of *G. biloba* for treating early-stage dementia in a community setting.	We found no evidence that a standard dose of high purity *G. biloba* confers benefit in mild to moderate dementia over 6 months.
[161]	*G. biloba* may have a role in treating impairments in memory, cognitive speed, activities of daily living (ADL), edema, inflammation, and free-radical toxicity associated with traumatic brain injury (TBI), Alzheimer’s dementia, stroke, vaso-occlusive disorders, and aging. The purpose of this review was to provide a synthesis of the mechanisms of action, clinical indications, and safety of *G. biloba* extract.	*Ginkgo* shows promise in treating some of the neurologic sequelae associated with Alzheimer’s disease, TBI, stroke, normal aging, edema, tinnitus, and macular degeneration. Mechanisms of action may include antioxidant, neurotransmitter/receptor modulatory, and antiplatelet-activating factor properties. While safe, caution is advised when recommending *Ginkgo* to patients taking anticoagulants. Future studies should examine dose effects, component activity, mechanisms, and clinical applications.
[162]	To assess the cumulative evidence on the efficacy and effectiveness of *G. biloba* extract (GbE) in the treatment of dementia.	GbE has potentially beneficial effects for people with dementia when it is administered at doses greater than 200 mg/day for at least 5 months. Given the lower quality of the evidence, further rigorously designed, multicenter-based, large-scale RCTs are warranted.
[163]	Prospective cohort studies showed inverse associations between the intake of flavonoid-rich foods (cocoa and tea) and cardiovascular disease (CVD). Intervention studies showed protective effects on intermediate markers of CVD. This may be due to the protective effects of the flavonoids epicatechin (in cocoa and tea) and quercetin (in tea) were investigated for the effects of supplementation of pure epicatechin and quercetin on vascular function and cardiometabolic health.	Our results suggest that epicatechin may partially contribute to the cardioprotective effects of cocoa and tea by improving insulin resistance. It is unlikely that quercetin plays an important role in the cardioprotective effects of tea. This study was registered at clinicaltrials.gov as NCT01691404.
[164]	This group studied the effects of supplementation of pure epicatechin and quercetin on biomarkers of endothelial dysfunction and inflammation.	In (pre)hypertensive men and women, epicatechin may contribute to the cardioprotective effects of cocoa and tea through improvements in endothelial function. Quercetin may contribute to the cardioprotective effects of tea possibly by improving endothelial function and reducing inflammation. This trial was registered at clinicaltrials.gov as NCT01691404.

## 9. Microbial Metabolism vs. Microbial Biotransformation and Engineering

Bacterial metabolism and biotransformation are distinct systems of molecular processing by microorganisms of sources such as plant material. Natural products and compounds with potential as phytoceuticals, or as drug candidates, mostly occur in nature at very low concentrations. Emerging alternatives to plant purification is to express biosynthetic genes in microbial systems such as bacteria and fungi [165]. More than 60% of antibiotics used in medical and other health practices are microorganism-derived natural products [166]. Primary metabolism centers around reactions associated with generating energy, biomass, and essential cellular components. Secondary metabolism, or xenobiotic metabolism, usually involves a non-survival role and adaptation to environmental conditions or offers some selective advantage. These compounds often have unusual structures and are normally synthesized later in the cell growth cycle and may have importance medicinally [167]. It is unquestioned that co-metabolism between the host and its resident microbiota produce naturally occurring secondary metabolites and any bioactivity owed to plant products must be considered against the backdrop of bacterial biotransformation. In order for proper evaluation and discovery, microbial biotransformation has emerged as an important player and a tool for obtaining novel structural drug analogs or to improve the pharmacokinetic parameters of other substances [168]. We have harnessed bacteria as programmable biochemical factories for diverse applications and have suggested the genetic engineering of bacteria for the production of valuable bioactive compounds through genome engineering CRISP-R and recombinant DNA technology [4]. These comprehensive engineering approaches are directed at improving the microbial production of natural products, proteins, and novel small molecules. Many of these bioactive compounds could be produced through engineering approaches to improve or augment the effectiveness or mechanism of action of *Ginkgo biloba*, as may be the case for the pathway engineering of primary bacterial metabolism [169].

Phenolic acids, polyphenols, and indigestible carbohydrates such as pectin, hemicellulose, and various polysaccharides are contained in many fruits and fruit juices. These dietary constituents support select microbial species and their colonization through co-metabolism. It is the co-metabolism through which the gut microbiota imparts many beneficial effects on health through health-promoting foods. The same premise holds for whole foods and *Ginkgo* or other so-called superfood constituents. For example, microbial fermentation by-products are SCFAs—in particular, acetate, propionate, and butyrate [170]. These prebiotics, in fact, act synergistically to modify colonic and intestinal microbiota, leading to a symbiosis with the human host that benefits health [171]. When fermented kiwifruit was added, SCFAs increased, while lactate and after 24 h succinate concentrations declined and most of the genera observed produced acetic acid [171]. Propionic acid is a SCFA product of *Bacteroidetes* fermentation and from *Veillonella*, members of the phyla *Firmicutes*. Butyrate, produced mainly by the *Firmicutes* subset, among others, features members of the *Lachnospiraceae* family [171]. 

Protective aspects of butyrate have been found, for example, from prebiotic potato starch with *Faecalibacterium prausnitzii* on *Clostridioides difficile* infection and damage [126]. Other studies found prebiotic effects, using 16S rRNA pyrosequencing of the microbial DNA prepared from stool co-incubated with select fruits when fermented with feces from 10 donors [171]. Additionally, in the aforementioned study by Juliet and colleagues, by-products of fermentation contributed to the first step in microbial colonization by modulating both microbial numbers and gut flora ecology through the adhesion of different bacteria to the gut wall [172]. Co-metabolism is a two-way street, and the same group found an immediate and long-lasting reduction in the abundance of all members of the *Proteobacteria* phyla as well as some members of the *Firmicutes* and *Bacteroidetes* phyla, which were originally present in the original fecal inoculum. Further, other abundances decreased by half, namely, *Ruminococcaceae*. In contrast, *Bacteroidaceae, Lachnospiraceae*, and *Veillonella* members of the *Firmicutes* phyla and *Coriobacteriaceae* of the Actinobacteria phyla increased over 24 h. The *Lachnospiraceae* levels were sustained for up to 48 h and *Coriobacteriaceae* and *Bifidobacteraceae Actinobacteria* increased [171]. The same holds for the constituent metabolic products derived from Ginkgo or other so-called superfoods and their whole food source.

## 10. Conclusions

New concepts and divergent views, differing from the dogmatic and classic understanding, represent a paradigm shift and potential for discovering novel treatments of diseases. Building on our previous work on the metabolic interaction between any host and any given organism at the small molecule level [99] and the involvement of the MGB axis when coupled with host co-metabolism, we have elucidated the common underpinnings of disease pathogenesis in AD and VaD and how natural compounds and the MGB hold promise to incentivize research and alternative treatments for this potentially treatable form of dementia. Moreover, a growing body of evidence suggests a common etiology for Alzheimer’s disease, cardiovascular disease, and VaD [17,142,173,174,175]. 

The use of *Ginkgo biloba* is not new in medicine as it has been used in China for centuries. For *Ginkgo biloba* to be fully accepted and integrated into clinical practice, its effectiveness must first be evidence-based. Further, we must also consider the second genome and vast metabolism of the microbiota–gut–brain axis in the metabolism of this compound to fully assess its effectiveness. Nevertheless, there remains a place for *Ginkgo biloba* in medicinal chemistry, especially when one considers bacterial co-metabolism. It only seems appropriate to offer GLE as an adjunct treatment in Western medicine as some clinical trials are demonstrating efficacy of *Ginkgo biloba* in the treatment of dementia symptoms. Unfortunately, biases remain that can hinder phytoceutical development and acceptance. Many pharmaceutical companies are funding clinical trials, which may compel researchers to anticipate obtaining positive data in support of funding for *Ginkgo biloba*. Patients educated about *Ginkgo biloba* can often demonstrate anecdotal faith in the drug, compelling them to believe in its success. This acceptance can hinder the evaluation of any improvement in function. Moreover, many pharmaceutical companies are funding these trials, which compels researchers to expect positive data in support of *Ginkgo biloba*. Whole *Ginkgo biloba longa* and its extracts as medicine and phytoceutical are gaining support and consideration with the power of clinically statistically significant results, low cost, low adverse reactions, and its prominent use in Eastern medicine for decades. Hopefully, the evidence and mechanistic understanding will garner GLE use in Western medicine. Moreover, we call for the concomitant exploration of co-metabolism for this drug with our gut microbiota as it relates to the microbiota–gut–brain axis [17]. Moving forward, there remains a place for *Ginkgo biloba* in medicinal chemistry and an opportunity that needs development for *Ginkgo biloba* to be accepted and integrated into clinical practice. 

Even though there are a substantial number of clinical trials available, the efficacy of these Ginkgo-derived compounds or extracts in many clinical trials is limited in scope. For many trials, the statistical power has been low. It is not easy to gather patients for studies, as many of the participants are elderly and rely heavily on family support. Finding a family unit that has time to contribute to the study is an obstacle. However, regardless of the obstacles, further research for alternative treatments for dementia is essential. Lack of research in this area, either because of the large time commitment needed and the cost of conducting trials [9], hampers substantial research data. Therefore, the amount of data accumulated thus far is not sufficient to provide statistically sound evidence, even though clinical trials are proving the efficacy of *Ginkgo biloba* in treatment of dementia symptoms. So, interest and further research and trials in this direction are warranted to help develop more information and evidence regarding its potential use in the treatment of dementia.

What can be concluded from the clinical trials discussed in this review is that the pathophysiology of vascular-associated dementia is among the many etiologies of AD that provide numerous targets for treatment. With the pharmacodynamic range of EGb761 discussed, EGb761 is still a viable treatment option for both Alzheimer’s disease and vascular dementia [72]. *Ginkgo biloba* extract is an important and popular dietary supplement, which has a role in the prevention of neurological damage and improved memory enhancement [176] of patients suffering from AD or VaD. The possibility of improving cognition, stabilizing and decreasing the decline of mental function in patients with dementia using the Ginkgolides and the flavonoid glycosides exists. We suggest that GLE can help affected individuals and inconclusive studies should be revisited, taking the MGB axis and microbial metabolism into account when studying natural products. However, little exploration has been undertaken against the backdrop of gut microbiota, also called the second brain and the second genome, within us. How the microbiota modulates the effects of natural products and the downstream chemical profile of bacteria-derived constituents is an underappreciated and potentially transforming aspect of these brain-modulating nutraceuticals or phytoceuticals. Revisiting the clinical trials and considering the nutritional aspects and microbial constitution of the elderly should shed light on the varied results found with natural products and prosaic foods for AD and VaD.

## Figures and Tables

**Figure 1 life-13-00041-f001:**
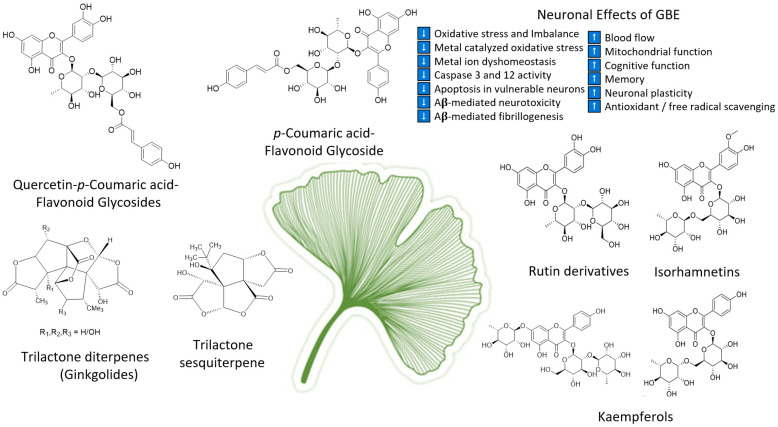
Structures of ginkgolides A, B, C (diterpenes), and bilobalide (sesquiterpene) and function of compounds found in *G. biloba* extract and EGb 761; proanthocyanidins, quercetins, coumarins, flavonoid glycosides, ginkgolides A, B, C-J, isorhamnetins, bilobalide, and rutin derivatives.

**Figure 2 life-13-00041-f002:**
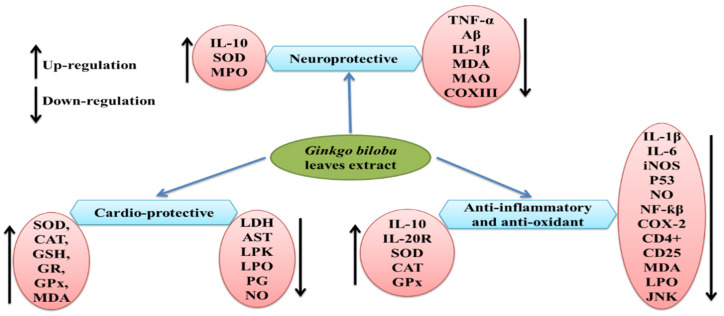
Regulation of various molecular targets and therapeutic effects of *Ginkgo biloba*. Aβ = amyloid beta; AST = aminotransferase; BCl-2 = B-cell lymphoma 2; CAT = catalase; CD4+ = cluster of differentiation 4; COX III = cyclooxygenase 3; GPx = glutathione peroxidase; GR = glutathione reductase; GSH = glutathione; IL-1β = interleukin 1 beta; IL-10 = interleukin 10; IL-20R = interleukin 20R; iNOS = nitric oxide synthase; JNK = c-Jun N-terminal kinase; LDH = lactate dehydrogenase; LPK = L-pyruvate kinase; LPO = lipid hydroperoxide; MAO = monoamino oxidase; MDA = malondialdehyde; MPO = myeloperoxidase; NO = nitric oxide; p53 = tumor protein p53; SOD = superoxide dismutase; TNF-α = tumor necrosis factor α.

## Data Availability

Not applicable.

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
