# Peer review of "Natural Product Co-Metabolism and the Microbiota–Gut–Brain Axis in Age-Related Diseases"

_life, 2022, doi:10.3390/life13010041_

Round 1
Reviewer 1 Report
In this review, the author discussed in detail how GLE co-metabolizes with GMB, and how it plays a role in the prevention and treatment of age-related diseases such as AD and VaD. In my opinion, this review is very comprehensive and enlightening, while a few issues should be resolved before its acceptance.
1. The language of this manuscript is satisfying, while many format errors should be addressed, such as L91, L104, and L176. The authors should re-check them carefully before its resubmission.
2. More related figures are recommending considering the extensive text information in this review. Table 1 should be in three-line format, and in Figure 1 the chemical structures of diterpenes and sesquiterpene are very similar. As far as I know, these two substances seem not have the same amount of isoprene, so I’m wondering if there would be an explanation.
3. Considering the medicinal value of GLE, advances related to the industrial preparation or synthetic biology of GLE are of great interest to me, and I wish the authors could further clarify this topic.
Reviewer 2 Report
The paper submitted by Obrenovich is a review focused on natural compounds and microbiota-gut axis in the context of age-related diseases. The topic is interesting, but the paper is not ready for publication at this stage as it should undergo major revision before further consideration. First, English language should be polished throughout the manuscript. Intended meaning of some sentences must be double checked, as well as the presence of typos and repetitions. Most importantly, some sections of the review may appear to the reader as “out of focus”, or at least not closely connected to the corresponding titles (check comments below).
· Title is not clear: the review appears to be more related to “metabolites” rather than “metabolism”. Moreover, since the paper is focused on G. biloba components, this could b clearly included in the title/abstract.
· Abstract should be improved. For example, repetitions should be avoided (e.g., “approaches” in the first sentence; “largely” page 3, line 127). Moreover, the meaning of some sentences is not clear (e.g. “particularly difficult to 23 treat let alone cure”).
· Figure 1: “neuronal affects” must be changed to “neuronal effects”. Resolution must be improved.
· Check references formatting: in some references, comma is missing (e.g. page 2, line 53).
· Page 3 line 121: check title of section 2, G. biloba should appear in italic
· Throughout the manuscript, authors should check how “G. biloba” has been written as it is not consistent (e.g. with or without capital letters, italics, abbreviated, etc…).
· I am concerned about the use of the definition of “therapeutic indications” for G. biloba in the context of AD treatment, since this is an experimental use of the extract (also, check cited reference 33).
· Section 2: the title could be revised as it may be misleading for the reader. In fact, the section is not only focused on G. biloba but more on biochemical pathways, in a more general fashion.
· Conclusion: the sentence “New concepts and divergent views, differing from the dogmatic or our classic understanding, represent a paradigm shift and potential for discovering novel disease pathways” is very general and may not be clear to the reader. Moreover, it does not add information to the manuscript.
Reviewer 3 Report
The manuscript “Natural Product Co-metabolism and the Microbiota-Gut-Brain Axis in Age-Related Diseases” fits the journal’s scope. The review is well-structured and comprehensively described, and the references are appropriate but the majority of them are not recent. The critical approach is sufficiently represented throughout the manuscript, being a major plus for the manuscript. However, the title is too general and can be misleading regarding the content of the manuscript. Thus, before publication, the authors should change the title according with the content of the manuscript.
Minor:
Line 139- please correct the error
Lines 21-23 – please rephrase
Line 24 – please correct the errors
Round 2
Reviewer 2 Report
The authors addressed my comments.